# Is Omalizumab Related to Ear and Labyrinth Disorders? A Disproportionality Analysis Based on a Global Pharmacovigilance Database

**DOI:** 10.3390/diagnostics12102434

**Published:** 2022-10-08

**Authors:** Hyeon Tae Park, Sunny Park, Yong Woo Jung, Soo An Choi

**Affiliations:** 1College of Pharmacy, Korea University, Sejong Campus, Sejong City 30019, Korea; 2College of Pharmacy and Research Institute of Pharmaceutical Sciences, Korea University, Sejong Campus, Sejong City 30019, Korea

**Keywords:** adverse event, omalizumab, eosinophilic otitis media, ear disorders, pharmacovigilance

## Abstract

Introduction: Asthma is a chronic disease, characterized by reversible airway obstruction, hypersensitivity reactions, and inflammation. Oral corticosteroids are an important treatment option for patients with severe or steroid-resistant asthma. Biologics for asthma are recommended in patients with severe asthma, owing to their steroid-sparing effect as well as their ability to reduce the severity and aggravation of uncontrolled asthma. Most clinical trials of omalizumab in patients with asthma have suggested its tolerability and safety. However, some studies reported eosinophilic comorbidities in the ear, nose, and throat during omalizumab treatment, particularly eosinophilic otitis media. This study examined the relationship between ear disorders and omalizumab compared with that of other biologics for asthma using a large real-world database. Materials and Methods: Individual case safety reports from the Uppsala Monitoring Centre Vigibase of biologics for asthma (omalizumab, mepolizumab, reslizumab, benralizumab, and dupilumab) up to 29 December 2019, were used. A disproportionality analysis was performed using the proportional reporting ratio (PRR), reporting odds ratio (ROR), and information components (IC). A hierarchy analysis used the Medical Dictionary for Regulatory Activities Terminology. A tree map was generated using R studio version 4.2. Results: In 32,618 omalizumab reports, 714 adverse events (AEs) were detected as signals. Among the 714 signals, seventeen AEs were detected as signals of omalizumab-related ear and labyrinth disorders in 394 reports. Only three AEs (ear pain, ear disorder, and ear discomfort) were detected from mepolizumab. No signal was detected from reslizumab, benralizumab, and dupilumab. Conclusions: Careful monitoring of ear disorders is recommended when omalizumab treatment is started, with decreased oral corticosteroid use in patients with severe asthma. Further studies are necessary to confirm the omalizumab-related signals.

## 1. Introduction

Asthma is a chronic disease characterized by reversible airway obstruction, hypersensitivity, and inflammation [1]. Inhaled corticosteroids (ICS) are generally used to manage asthma, and oral corticosteroids (OCS) are an important treatment for patients with severe or steroid-resistant asthma [2,3]. Severe asthma is defined as asthma requiring high doses of ICS, plus a second controller with or without systemic corticosteroids [4]. Patients with severe asthma experience increased hospitalization, poor quality of life, impaired lifestyle, and harmful OCS side effects [5]. Steroid use can lead to many adverse effects, including hyperglycemia, osteoporosis, and cataracts [6]. Due to the various adverse events (AEs) and the burden of chronic OCS use, considering the overall patient’s condition may be required [6]. Several biologics are currently available for patients with severe asthma.

Biologics for asthma target the specific inflammatory pathways involved in its pathogenesis [7]. Omalizumab, a recombinant humanized monoclonal antibody that inhibits IgE-mediated inflammation, was first approved by the Food and Drug Administration in 2003 for patients with asthma [3,8,9]. It binds to free IgE and prevents its attachment to mast cells and basophils, resulting in reduced histamine release [10]. Other biologics for asthma have also been developed after omalizumab, such as mepolizumab, reslizumab, benralizumab, and dupilumab. Mepolizumab, reslizumab, and benralizumab block IL-5 signaling and inhibit eosinophil proliferation and activation [11,12]. Dupilumab blocks both IL-4 and IL-13 signaling pathways [13]. Biologics have a steroid-sparing effect by reducing the severity and exacerbation of uncontrolled asthma, and are therefore recommended for patients with severe asthma [2,6]. Biologics for asthma showed their efficacy in reducing 59% of significant exacerbations, 65% of severe exacerbations, and 54% of maintenance OCS doses in 12 months [14].

Most clinical trials of omalizumab in patients with asthma have suggested its tolerability and safety [15,16,17]. However, a randomized clinical trial (RCT) of omalizumab in children with asthma reported earache [16]. Other studies reported that eosinophilic comorbidities could occur during omalizumab treatment in the ear, nose, and throat (ENT), particularly eosinophilic otitis media (EOM) [18]. EOM showed a high rate of comorbidity with asthma and is characterized by discharge with a high concentration of eosinophils and high viscosity in the middle ear cavity [19,20]. Only patients with moderate or severe persistent asthma showed EOM onset, with frequent incidences occurring in patients with severe asthma [20]. For EOM patients, systemic or topical steroid administration is considered the most effective option for asthma [21]. However, little is known about EOM, particularly following the initiation of biologics.

Although RCTs are the gold standard for verifying the efficacy of drugs before approval, AEs cannot be fully determined during drug development. The limitations of clinical trials include homogenous groups, small sample sizes (<1000 patients), a short duration, and an inability to anticipate real-world responses [22,23]. Pharmacovigilance (PV) data can provide meaningful insights into drugs in the real world [22]. Recent post-marketing studies suggested the risk of cancer [24] and anaphylactic shock [25] associated with omalizumab. Although EOM associated with omalizumab use has been reported, a clear understanding of comorbidity in asthma has not been elucidated. Thus, we examined the relationship between ear disorders and omalizumab compared to that for other biologics used to treat asthma using a large real-world database.

## 2. Materials and Methods

### 2.1. Data Source

This study used individual case safety reports (ICSRs) from the Uppsala Monitoring Centre Vigibase of biologics for asthma (omalizumab, mepolizumab, reslizumab, benralizumab, and dupilumab). Information contained in the database was reported by members participating in the WHO International Drug Monitoring program from 1968 until 29 December 2019. Data originated from local physicians, pharmacists, other healthcare providers, and the public. ICSRs contained information about the primary source, age, sex, name of the drug used, indication, and adverse events. We analyzed the reported data in total and omalizumab ear-related, including demographics, age, sex, disease seriousness, and notifiers. Serious AEs were defined as death, life-threatening, hospitalization (initial or prolonged), disability or permanent damage, congenital anomaly/birth defect, requiring intervention to prevent permanent impairment or damage, or other [26,27].

### 2.2. Data Mining and Signal Detection Criteria

A two-by-two table was used to analyze the disproportionality (Table 1), a method that has been used as the basic approach for detecting signals in large databases [28]. We calculated the proportional reporting ratio (PRR), reporting odds ratio (ROR), and information component (IC), which are the most frequently used disproportionality parameters [29,30]. For events reported at least three times, signals were defined when the PRR and ROR were above 2 and below the IC limit of 95% above 0 (Table 2). Omalizumab-related signals were detected and compared to those of reslizumab, mepolizumab, benralizumab, and dupilumab.

Eosinophil-related signals were extracted from biologics for asthma to clarify the relationship between eosinophils and ear disorders. Among the detected signals, eosinophil-related signals were defined as AEs with “eosinophils” in the preferred terms.

### 2.3. Hierarchy Analysis

Medical Dictionary for Regulatory Activities (MedDRA) terminology, the global standard hierarchy for recording AEs and medical history, was adopted [31]. It has five hierarchical sub-categories: system organ class (SOC), high-level group term (HLGT), high-level term (HLT), preferred term (PT), and lowest-level term (LLT) [32]. MedDRA is a multiaxial terminology that refers to the representation of a medical concept in multiple SOCs. In this study, only primary SOCs were used to analyze the hierarchy. PTs of MedDRA version 23.0 were used. A hierarchy analysis was performed to determine the overall safety profiles. Hierarchical data were visualized by tree mapping using R Studio ver. 4.1.2.

## 3. Results

### 3.1. Characteristics of Omalizumab-Related AE Reports

The demographics of omalizumab-related AE reports are presented in Table 3. There were 32,618 omalizumab AE reports and 586 reports for ear and labyrinth disorders. Approximately 80% of omalizumab AE reports were from the Americas. AEs related to ear and labyrinth were mostly from the Americas (87%). Females were more likely to report AEs of omalizumab than males. The most frequently reported age was 45–65 years. Of all omalizumab reports, 16,592 (50.87%) were reported as serious. Among 586 reports of omalizumab-related ear and labyrinth disorders, 393 (67.06 %) were reported as serious. Approximately two-thirds of omalizumab reports were from professionals, including physicians, pharmacists, and other healthcare professionals.

### 3.2. Detected Signals of Omalizumab in Terms of System Organ Class (SOC)

In the disproportionality analysis, 714 AEs were detected as signals. The omalizumab signals were visualized using hierarchical analysis, as shown in Figure 1. The size of the treemap expresses the number of signals, and the color represents the number of reports. AEs in respiratory, thoracic, and mediastinal disorders were the most frequently reported as being related to asthma symptoms.

#### 3.2.1. Signals for Ear and Labyrinth Disorders

Seventeen AEs were detected as signals of omalizumab-related ear and labyrinth disorders in 394 reports, whereas only three AEs (ear pain, ear disorder, and ear discomfort) were detected for mepolizumab, and no signal was detected for reslizumab, benralizumab, and dupilumab (Table 4). The number of reports of ear pain, ear disorder, and ear discomfort with mepolizumab was 19, 8, and 8, respectively. Table 5 shows the disproportionality analysis results for omalizumab-related ear and labyrinth disorders. Ear pain was the most frequently reported ear-related AE of omalizumab, followed by ear discomfort and otorrhoea.

#### 3.2.2. Eosinophil-Related Signals for Omalizumab

Eight eosinophil-related AE signals were detected in omalizumab, as shown in Table 6. Eosinophilic granulomatosis with polyangiitis was omalizumab’s most frequently reported AE (124 reports). Other eosinophil disorders, including eosinophilic pneumonia and esophagitis, were also detected. With benralizumab and dupilumab, abnormal eosinophil counts were detected in 11 and 5 reports, respectively. In mepolizumab, eosinophilic bronchitis and eosinophil esophagitis were detected as signals in four reports each.

## 4. Discussion

This study revealed omalizumab-related ear disorders using a real-world pharmacovigilance database. Although omalizumab safety was generally considered tolerable, as reported in clinical trials [15,16], a RCT [16] and a report [18] demonstrated the onset of ear disorders after omalizumab use in patients with asthma. This study found symptoms of EOM, such as deafness, otorrhea, and tympanic membrane perforation, in omalizumab AE reports, consistent with previous studies [20,21,33,34]. In addition, increased eosinophil effects were detected following omalizumab treatment. In contrast, few signals related to the ear or eosinophils were detected in other biologics for asthma.

Although it is possible that omalizumab itself can lead to ear disorders, a focus on researching the disease and treatment background is needed, because EOM is associated with asthma, and EOM onset was only observed in patients with severe persistent asthma [20]. In this study, the serious AE percentage was relatively higher than the 13.6% that was previously shown for montelukast [35]. These results indicate that biologics, including omalizumab, may be relevant to EOM comorbidity in patients under severe disease conditions. Comorbid EOM was managed well during the prevalent use of OCS for asthma treatment [36]. Corticosteroids inhibit several pro-inflammatory cytokines and the chemokines associated with most immune cells [37]. However, after starting biologics, OCS use could be reduced to minimize their side effects. Therefore, the steroid-sparing effects of biologics for asthma may mask comorbid ear symptoms in patients with severe asthma.

In fact, many case reports suggested the successful treatment of EOM with mepolizumab [38], benralizumab [39], and omalizumab [40]. Our study confirmed the ear-related signals of omalizumab, whereas no such signal was detected for other biologics in asthma. It seems that the biologics for asthma did not cause the same results in ear disorders. Despite the similar chemical nature of biologics, differences in the biologics’ mechanisms of action may explain the different results. Anti-IL5/5R agents, including mepolizumab, reslizumab, and benralizumab, suppress the growth, differentiation, and proliferation of eosinophils and mast cells [14,21,41,42,43]. Dupilumab, an anti-IL4R agent, has the same effect on immune cells as that of anti-IL5/5R agents by inhibiting the IL-13 signaling pathway [44,45] and suppressing the release of IgE from B cells by blocking the IL-4 signaling pathway [45,46]. Mast cells cause allergic reactions through IgE-dependent and-independent processes [47,48,49]. Mepolizumab, reslizumab, benralizumab, and dupilumab appear to reduce mast cell degranulation and allergic reactions by inhibiting both IgE-dependent and independent pathways [41,42,44,46,47,48,50]. In contrast, omalizumab only blocks the binding between IgE and FceRI [51] and cannot reduce inflammation by IgE-independent degranulation, although omalizumab can inhibit inflammation by blocking the IgE-dependent degranulation of mast cells. Therefore, omalizumab does not seem to be able to completely control eosinophil level only by inhibiting the IgE-dependent pathway, leading to the aggravation of eosinophil-related disorders.

Similar to ear-related signals, our disproportionality results found signals related to the increased eosinophil counts of omalizumab, whereas other biologics for asthma showed relatively little association with eosinophils. As EOM is characterized by the accumulation of eosinophils in middle ear effusion and middle ear mucosa [45], signals for the increased eosinophils of omalizumab, unlike other biologics, could support omalizumab-related ear disorders. However, the effects of omalizumab on eosinophil levels remain controversial. A study revealed no decrease in the eosinophil count during omalizumab treatment, despite a reduction in asthma exacerbations, hospitalization, and an improvement in forced expiratory volume at 1 s [52]. In contrast, a previous study also reported reduced blood eosinophil numbers after omalizumab use [43]. A pilot study showed that omalizumab-induced remission in eosinophilic esophagitis patients was limited to patients with a low eosinophil count [53]. Racial differences were reported in patients, with eosinophilic esophagitis being more common in Caucasians than African Americans [54]. However, this study could not evaluate racial differences in overall and eosinophilic related omalizumab disorders, because most of the ICSRs were from America and Europe (96%).

This study has limitations in that the PV database has the possibility of over- and under-reporting biases, and the study cannot prove a causal relationship between drugs and AEs. In addition, changes in OCS use in ear-disorder reports or pharmacokinetic/pharmacodynamics analysis could not be examined due to insufficient data. Therefore, further studies including a controlled trial, cohort, or case-control study are needed to confirm omalizumab-related signals. Nevertheless, this study is valuable in several ways, showing the possible pathology of ear disorders after omalizumab use based on real-world data, including post-marketing AE reports, contributing to a more precise diagnosis of ear disorder events. The results imply that careful monitoring of EOM is needed when omalizumab is initiated with decreasing OCS use. Furthermore, our study shows a new paradigm for PV database utilization, suggesting that treatment precautions are not always limited to the drug–AE relationship. Despite the small sample size and short duration, several studies have described the use of omalizumab to treat EOM [40,55]. However, the effects of omalizumab at the eosinophil level seemed to be controversial and incomplete in that it inhibits only the IgE-dependent pathway. In addition, our study showed increased eosinophil levels and ear disorders after omalizumab use. Therefore, the application of omalizumab to EOM should be careful and eosinophil-related disorders should be monitored when omalizumab starts, particularly with reduced OCS use.

## 5. Conclusions

Based on a real-world global database, the careful monitoring of ear disorders is recommended when omalizumab treatment is initiated, with decreased oral corticosteroid use in patients with severe asthma. Further studies are necessary to confirm the omalizumab-related signals.

## Figures and Tables

**Figure 1 diagnostics-12-02434-f001:**
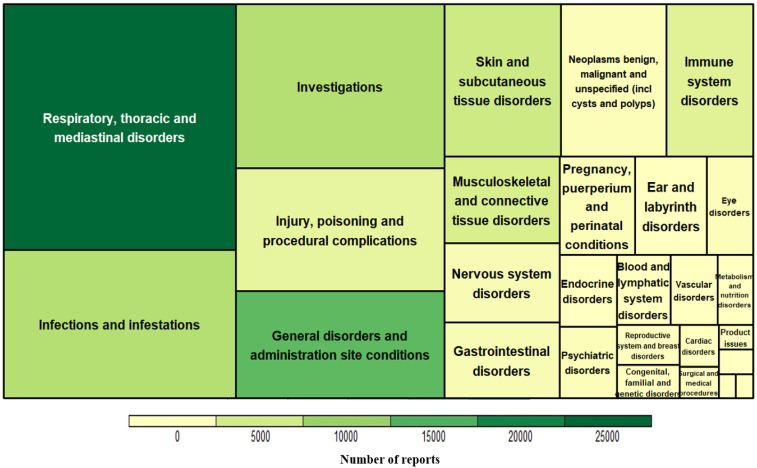
Treemap of signals for omalizumab by system organ class.

**Table 1 diagnostics-12-02434-t001:** Two-by-two contingency table for disproportionality analysis.

Number of Reports	Specific AEs	All other AEs
Target drug	A	B
All other drugs	C	D

The number of reports contained in: A: both target drugs and specific adverse events; B: target drug adverse events but with all other adverse events; C: specific adverse events but with all other drugs; D: all other drugs and all other adverse events.

**Table 2 diagnostics-12-02434-t002:** Formulae and criteria for signal detection.

Indices	Formula	Criteria
PRR	[A/(A + B)]/[C/(C + D)]	PRR ≥ 2
ROR	(A/B)/(C/D)	ROR ≥ 2
IC	IC = log_2_P(AE, Drug)/P(AE)P(Drug)	Under a limit of 95% IC ≥ 0

RRR, proportional reporting ratio; ROR, reporting odds ratio; IC, information component; AE, adverse event; CI, confidence interval.

**Table 3 diagnostics-12-02434-t003:** Demographic data of omalizumab on Vigibase.

Demographics	No. of Reports of Omalizumab (%) [N = 32,618]	No. of Reports of Omalizumab Related to Ear and Labyrinth Disorders (%) [N = 586]
Continent		
Africa	142 (0.44%)	1 (0.17%)
Americas	25,516 (78.23%)	512 (87.37%)
Asia	1321 (4.05%)	5 (0.85%)
Europe	5399 (16.55%)	67 (11.43%)
Oceania	240 (0.74%)	1 (0.17%)
Age group		
0–27 days	37 (0.11%)	1 (0.17%)
28 days to 23 months	36 (0.11%)	0 (0%)
2–11 years	425 (1.3%)	3 (0.51%)
12–17 years	1068 (3.27%)	20 (0.341%)
18–44 years	5712 (17.51%)	134 (22.87%)
45–65 years	7192 (22.05%)	192 (32.76%)
65–74 years	1963 (6.02%)	47 (8.02%)
≥75 years	838 (2.57%)	20 (3.41%)
Unknown	15,347 (47.05%)	169 (28.84%)
Gender		
Male	8880 (27.22%)	183 (31.23%)
Female	21,078 (64.62%)	393 (67.06%)
Unknown	460 (8.16%)	10 (1.71%)
Serious		
Yes	16,592 (50.87%)	393 (67.06%)
No	15,330 (47.00%)	186 (31.74%)
Unknown	696 (2.13%)	7 (1.19%)
Notifier		
Physician	14,137 (43.34%)	230 (39.25%)
Pharmacist	1057 (3.24%)	7 (1.19%)
Other Health Professional	6383 (19.57%)	121 (20.65%)
Consumer/Non-Health Professional	9773 (29.96%)	195 (33.28%)
Lawyer	8 (0.02%)	1 (0.17%)
Unknown	1260 (3.86)	32 (5.46%)

**Table 4 diagnostics-12-02434-t004:** Number of reports and signals for ear and labyrinth disorders.

Agents	Total Number of Reports	Number of Signals for Ear and Labyrinth Disorders	Number of Reports for Ear and Labyrinth Disorder Signals
Omalizumab	32,618	17	394
Mepolizumab	7344	3	35
Benralizumab	2387	0	0
Reslizumab	315	0	0
Dupilumab	20,559	0	0

**Table 5 diagnostics-12-02434-t005:** Disproportionality analysis results of biologics for asthma related ear and labyrinth disorders.

AE	No. of Reports (%)	PRR	ROR	IC_025_
Omalizumab				
Ear pain	152 (38.58%)	6.27	6.30	2.37
Ear discomfort	76 (19.29%)	7.10	7.12	2.42
Otorrhoea	25 (6.35%)	17.17	17.18	3.06
Ear pruritus	23 (5.84%)	8.22	8.23	2.17
Ear congestion	21 (5.33%)	8.55	8.56	2.17
Ear swelling	17 (4.31%)	7.17	7.17	1.83
Ear disorder	16 (4.06%)	2.30	2.31	0.35
Tympanic membrane perforation	12 (3.05%)	7.77	7.78	1.67
Deafness unilateral	11 (2.79%)	2.19	2.19	0.08
Vertigo positional	8 (2.03%)	4.59	4.59	0.75
Middle ear effusion	8 (2.03%)	3.90	3.90	0.56
Meniere’s disease	7 (1.78%)	4.01	4.01	0.47
Cerumen impaction	5 (1.27%)	4.71	4.71	0.28
Tympanic membrane disorder	4 (1.02%)	10.72	10.72	0.62
Tympanic membrane scarring	3 (0.76%)	162.74	162.76	0.69
Eustachian tube obstruction	3 (0.76%)	17.75	17.76	0.33
Eustachian tube disorder	3 (0.76%)	13.11	13.11	0.21
Mepolizumab				
Ear pain	19 (54.2%)	3.47	3.47	0.98
Ear disorder	8 (22.9%)	5.13	5.14	0.88
Ear discomfort	8 (22.9%)	3.30	3.31	0.37

IC_025_ under a limit of 95% IC.

**Table 6 diagnostics-12-02434-t006:** Disproportionality analysis results of eosinophil-related AEs of biologics for asthma.

AE	No. of Reports	PRR	ROR	IC_025_
Omalizumab				
Eosinophilic granulomatosis with polyangiitis	124	81.54	81.85	5.82
Eosinophilia	77	2.35	2.36	0.89
Eosinophil count increase	76	12.29	12.31	3.15
Eosinophilic pneumonia	14	6.00	6.00	2.35
Eosinophilic oesophagitis	11	27.75	27.76	3.66
Hypereosinophilic syndrome	8	41.66	41.67	2.43
Eosinophilic pneumonia chronic	5	50.86	50.87	3.18
Eosinophil count abnormal	3	14.47	14.47	0.25
Mepolizumab				
Eosinophilic bronchitis	4	726.14	726.54	1.41
Eosinophilic oesophagitis	4	43.84	43.87	1.19
Benralizumab				
Eosinophil count abnormal	10	775.56	779.19	4.48
Dupilumab				
Eosinophil count abnormal	5	38.68	38.68	1.59

IC_025_ under a limit of 95% IC.

## Data Availability

Our study used UMC Vigibase, and the database’s transfer, rent, or sale to any third party other than researchers was forbidden. Data will be available after approval is obtained from the UMC at https://who-umc.org/ (accessed on 1 September 2022) (request number ER198-2019).

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
