# Peer review of "Is Omalizumab Related to Ear and Labyrinth Disorders? A Disproportionality Analysis Based on a Global Pharmacovigilance Database"

_diagnostics, 2022, doi:10.3390/diagnostics12102434_

Round 1

Reviewer 1 Report

Withdrawal of systemic corticosteroids could be the cause of many of the EOS-related adverse events. In other words, EGPA and many other eosinophilic syndromes were masked by SCS and otitis media.

There is a great difference in the time since launching the biologics agents. For example, omalizumab has more than 15 years in the market; while dupilumab has less than a decade. How can the authors balance this issue?

Reference 2 might include the date of the consult. GINA 2022?

Author Response

Withdrawal of systemic corticosteroids could be the cause of many of the EOS-related adverse events. In other words, EGPA and many other eosinophilic syndromes were masked by SCS and otitis media.

There is a great difference in the time since launching the biologics agents. For example, omalizumab has more than 15 years in the market; while dupilumab has less than a decade. How can the authors balance this issue?

→ Signal detection is to find out signals using disproportionality of interest AE comparing to all AEs. It assumes that all AEs and drugs, not limited to specific AEs or drugs, were reported reflecting real-world occurrences of AEs. Even though launching time and the number of reports were different among biologics, disproportionality analysis is availabe because all AEs and drugs were assumed to be reported with similar reporting rate irrespective of drugs or the number of reports. Therefore, this study could find and suggest significant signals using disproportionality analysis. Additionally, there were not many reports from mepolizumab, benralizumab, and reslizumab, however, omalizumab and dupilumab have enough safety reports to conduct disproportionality analysis in this study.

 Reference 2 might include the date of the consult. GINA 2022?

→ As your comment, the date of guideline (reference 2) was added.

Reviewer 2 Report

Although the manuscript is well written and the work is highly organized, you need to improve the discussion of the results based on both the pharmacodynamic and pharmacokinetic bases.

Author Response

Although the manuscript is well written and the work is highly organized, you need to improve the discussion of the results based on both the pharmacodynamic and pharmacokinetic bases.

→ Our study used Vigibase (global database of individual case safety reports) and information is reporter dependent. Therefore, dosing information and other lab values including IgE or eosinophil level were not enough to determine PK and PD from our data. Therefore, we added that limitation in the final paragraph of discussion section.

Reviewer 3 Report

This research aims to investigate the relation between ear disorders and omalizumab in comparison to other biologists used to treat asthma.

The paper is interesting. Below are a few suggestions to enhance the paper's quality:

1. Replace Figure 1 with a better resolution format.

2. Revise the discussion section to describe how the chemical nature and mode of action of various biologics used to treat asthma result in differences in the prevalence of adverse drug reactions.

3. in the discussion section, give a paragraph to describe the potential influence of different racial groups on the emergence of adverse medication responses.

Author Response

  1. Replace Figure 1 with a better resolution format.

→ New version of Figure 1 was added. The different ratio of width and length made the different arrangement of figure.

  1. Revise the discussion section to describe how the chemical nature and mode of action of various biologics used to treat asthma result in differences in the prevalence of adverse drug reactions.

→ As you mentioned, 5 biologics have the different mode of actions leading to the different safety profiles of adverse drug reactions, particular in eosinophil-related disorders. It was discussed in the 3rd paragraph of discussion section with additional expression.

  1. in the discussion section, give a paragraph to describe the potential influence of different racial groups on the emergence of adverse medication responses.

→ There are previous studies showing racial difference in eosinophilic disorders, however, this study could not evaluate because most of the individual safety reports of omalizumab were from America and Europe (96%). Therefore, we described additional comments about racial difference in the 4th paragraph of discussion section.

Reviewer 4 Report

This is an interesting paper, assessing the relationship between ear and labyrinth disorders and omalizumab compared with other biological medicines used for asthma treatment. This study used individual case safety reports (ICSRs) from the Uppsala Monitoring Centre Vigibase. The methodology was based on signal detection using the proportional reporting ratio (PRR), reporting odds ratio (ROR), and information components (IC).

Major comments

1) However, it’s important to note that a signal does not indicate a direct causal relationship between a side effect and a medicine, but is essentially only a hypothesis that, together with data and arguments, justifies the need for further assessment. For this reason, it should be highlighted in the conclusion of the abstract of the manuscript, and in the discussion section that other additional studies (such as case-control, cohort studies, etc) are necessary to confirm the detected signals.

2) Although the abstract indicates the study examined the relationship between ear or labyrinth disorders and the biological drugs used in asthma, the study also evaluated eosinophil-related disorders. Even though in smaller numbers signals related to the ear and labyrinth or eosinophils were also detected with other biologics for asthma (ear signals with mepolizumab, eosinophilic-related adverse events with mepolizumab, benralizumab and dupilumab).

Minor comments

11) The database source of analysis should be reported in the material and methods section of the abstract.

12) In line 110 the term PTs appears for the first time without specifying the meaning. Although later in line 115 PT is indicated as the preferred term.

Author Response

Major comments

However, it’s important to note that a signal does not indicate a direct causal relationship between a side effect and a medicine, but is essentially only a hypothesis that, together with data and arguments, justifies the need for further assessment. For this reason, it should be highlighted in the conclusion of the abstract of the manuscript, and in the discussion section that other additional studies (such as case-control, cohort studies, etc) are necessary to confirm the detected signals.

→ Thank you for your comments. The necessity of further study was added in the conclusion of the abstract and discussion section of manuscript.

Although the abstract indicates the study examined the relationship between ear or labyrinth disorders and the biological drugs used in asthma, the study also evaluated eosinophil-related disorders. Even though in smaller numbers signals related to the ear and labyrinth or eosinophils were also detected with other biologics for asthma (ear signals with mepolizumab, eosinophilic-related adverse events with mepolizumab, benralizumab and dupilumab).

→ We agree with your points. Comparing to other biologics, various ear and eosinophil-related signals were detected with omalizumab. In particular, “eosinophil count abnormal” with benralizumab and dupilumab shows “abnormal” and it could not be confirmed increase or decrease. In case of “eosinophilic bronchitis” with mepolizumab, it could be due to background disease. In contrast, clear signals of increased eosinophil with omalizumab (including eosinophilic granulomatosis with polyangiitis, eosinophilia, eosinophil count increase or hypereosinophilic syndrome) were detected. These points were described in the 1st paragragh of discussion section.

Minor comments

The database source of analysis should be reported in the material and methods section of the abstract.

→ We added database source (Vigibase) in materials and methods section of abstract.

In line 110 the term PTs appears for the first time without specifying the meaning. Although later in line 115 PT is indicated as the preferred term.

→ “PTs” in line 110 was corrected to “preferred terms”.

Round 2

Reviewer 1 Report

nothing to comment

Author Response

Thank you for your reply.